# C-Reactive Protein (CRP) Levels in Immune Checkpoint Inhibitor Response and Progression in Advanced Non-Small Cell Lung Cancer: A Bi-Center Study

**DOI:** 10.3390/cancers12082319

**Published:** 2020-08-17

**Authors:** Jakob M. Riedl, Dominik A. Barth, Wolfgang M. Brueckl, Gloria Zeitler, Vasile Foris, Stefanie Mollnar, Michael Stotz, Christopher H. Rossmann, Angelika Terbuch, Marija Balic, Tobias Niedrist, Thomas Bertsch, Herbert Stoeger, Martin Pichler, Horst Olschewski, Gudrun Absenger, Joachim H. Ficker, Armin Gerger, Florian Posch

**Affiliations:** 1Division of Oncology, Department of Internal Medicine, Medical University of Graz, Auenbruggerplatz 15, A-8036 Graz, Austria; j.riedl@medunigraz.at (J.M.R.); dominik.barth@medunigraz.at (D.A.B.); stefanie.mollnar@stud.medunigraz.at (S.M.); michael.stotz@medunigraz.at (M.S.); christopherherbert.rossmann@klinikum-graz.at (C.H.R.); angelika.terbuch@medunigraz.at (A.T.); marija.balic@medunigraz.at (M.B.); herbert.stoeger@medunigraz.at (H.S.); martin.pichler@medunigraz.at (M.P.); gudrun.absenger@medunigraz.at (G.A.); armin.gerger@medunigraz.at (A.G.); 2Department of Translational Molecular Pathology, MD Anderson Cancer Center, Houston, TX 77030, USA; 3Department of Respiratory Medicine, Allergology and Sleep Medicine/Nuremberg Lung Cancer Center, Nuremberg General Hospital, Prof.-Ernst-Nathan-Str. 1, 90419 Nuremberg, Germany; wolfgang.brueckl@klinikum-nuernberg.de (W.M.B.); joachim.ficker@klinikum-nuernberg.de (J.H.F.); 4Department of Respiratory Medicine, Paracelsus Medical Private University Nuremberg, Prof.-Ernst-Nathan-Str. 1, 90419 Nuremberg, Germany; Gloria.Zeitler@stud.pmu.ac.at (G.Z.); thomas.bertsch@klinikum-nuernberg.de (T.B.); 5Division of Pulmonology, Department of Internal Medicine, Medical University of Graz, Auenbruggerplatz 15, A-8036 Graz, Austria; vasile.foris@medunigraz.at (V.F.); horst.olschewski@medunigraz.at (H.O.); 6Ludwig Boltzmann Institute for Lung Vascular Research, A-8010 Graz, Austria; 7Clinical Institute of Medical and Chemical Laboratory Diagnostics, Medical University of Graz, Auenbruggerplatz 15, A-8036 Graz, Austria; tobias.niedrist@medunigraz.at; 8Institute of Clinical Chemistry, Laboratory Medicine and Transfusion Medicine Nuremberg General Hospital, Prof.-Ernst-Nathan-Str., 90419 Nuremberg, Germany; 9Department of Experimental Therapeutics, The University of Texas MD Anderson Cancer Center, Houston, TX 77054, USA; 10Center for Biomarker Research in Medicine (CBmed Ges.m.b.H.), Stiftingtalstrasse 5, A-8010 Graz, Austria

**Keywords:** C-reactive protein, immune checkpoint inhibition, biomarker, treatment response, joint model

## Abstract

Background: Biomarkers for predicting response to immune checkpoint inhibitors (ICI) are scarce and often lack external validation. This study provides a comprehensive investigation of pretreatment C-reactive protein (CRP) levels as well as its longitudinal trajectories as a marker of treatment response and disease outcome in patients with advanced non-small cell lung cancer (NSCLC) undergoing immunotherapy with anti PD-1 or anti PD-L1 agents. Methods: We performed a retrospective bi-center study to assess the association between baseline CRP levels and anti PD-(L)1 treatment outcomes in the discovery cohort (*n* = 90), confirm these findings in an external validation cohort (*n* = 101) and explore the longitudinal evolution of CRP during anti PD-(L)1 treatment and the potential impact of dynamic CRP changes on treatment response and disease outcome in the discovery cohort. Joint models were implemented to evaluate the association of longitudinal CRP trajectories and progression risk. Primary treatment outcomes were progression-free survival (PFS) and overall survival (OS), while the objective response rate (ORR) was a secondary outcome, respectively. Results: In the discovery cohort, elevated pretreatment CRP levels emerged as independent predictors of worse PFS (HR per doubling of baseline CRP = 1.37, 95% CI: 1.16–1.63, *p* < 0.0001), worse OS (HR per doubling of baseline CRP = 1.42, 95% CI: 1.18–1.71, *p* < 0.0001) and a lower ORR ((odds ratio (OR) of ORR per doubling of baseline CRP = 0.68, 95% CI: 0.51–0.92, *p* = 0.013)). In the validation cohort, pretreatment CRP could be fully confirmed as a predictor of PFS and OS, but not ORR. Elevated trajectories of CRP during anti PD-(L)1 treatment (adjusted HR per 10 mg/L increase in CRP = 1.22, 95% CI: 1.15–1.30, *p* < 0.0001), as well as a faster increases of CRP over time (HR per 10 mg/L/month faster increase in CRP levels = 13.26, 95% CI: 1.14–154.54, *p* = 0.039) were strong predictors of an elevated progression risk, whereas an early decline of CRP was significantly associated with a reduction in PFS risk (HR = 0.91, 95% CI: 0.83–0.99, *p* = 0.036), respectively. Conclusion: These findings support the concept that CRP should be further explored by future prospective studies as a simple non-invasive biomarker for assessing treatment benefit during anti PD-(L)1 treatment in advanced NSCLC.

## 1. Introduction

In 2015 two groundbreaking phase III trials demonstrated superiority of the anti PD-1 inhibitor nivolumab over standard chemotherapy with docetaxel in the second line treatment of non-small cell lung cancer [1,2]. Subsequently, the PD-1 inhibitor pembrolizumab and the PD-L1 inhibitor Atezolizumab showed comparable efficacy, leading to a paradigm shift in the treatment of advanced NSCLC [3,4]. Today anti PD-(L)1 agents, either as monotherapy or as combination with platinum based chemotherapy, represent the mainstay of palliative first line treatment in advanced NSCLC without activating mutations. These novel agents improved response rates, prolonged survival and achieved a higher percentage of durable responses with less side effects compared to chemotherapy alone [5,6,7,8,9].

However, a major proportion of NSCLC patients treated with immune checkpoint inhibitors (ICI) are primary nonresponders, and more than two thirds of patients develop treatment resistance over time [10]. Thus, there is an urgent need for novel readily available and reproducible predictive biomarkers in order to improve patient selection, maximize treatment benefit, minimize treatment side effects and avoid unnecessary costs of ICI treatment. To date, PD-L1 expression on cancer cells determined by immunohistochemistry and the tumor mutational burden assessed by next-generation sequencing (NGS) are the only validated predictive markers for ICI response in randomized phase III trials, of which at least the latter lacks standardization [11,12].

Elevated levels of baseline C-reactive protein (CRP), a marker of systemic inflammation and immune activation, are associated with poor treatment response to chemotherapy and adverse disease outcome in various cancers including advanced NSCLC [13,14,15,16]. We can conceive a strong biologic rationale on how elevated CRP may be a proxy marker for adverse immunotherapy outcomes and disease progression in NSCLC. Cancer-related inflammation in the tumor microenvironment fosters cancer progression by promoting cell proliferation, angiogenesis and cancer cell migration. In addition to the local inflammatory response cancer cells mediate systemic inflammation which is orchestrated by various subtypes of immune cells, cytokines and acute phase proteins [17,18]. Cancer-induced systemic inflammation often marks cancer progression in clinical medicine and is thought to contribute to various cancer-related complications such as cachexia, pyrexia and fatigue. CRP, as an acute phase protein of hepatic origin, has been shown to reflect this process [17,19,20,21,22,23,24,25]. Moreover, systemic inflammation as reflected by CRP has been suggested to correlate with low levels of CD4+ T-cells, which play a key role in ICI mediated antitumor immune response [26].

Importantly, recent studies have indicated that pretreatment CRP may represent a valuable prognostic marker in the immunotherapy setting of advanced NSCLC [27,28,29]. However, these studies were either limited by a very small sample size or the lack of an external validation. In addition, cancer is a highly dynamic disease, and its course within an individual patient is strongly influenced by various external and internal factors, including treatment interventions, cancer evolution and acquired resistance mechanisms to antineoplastic therapy [30]. We hypothesized that longitudinal measurements of CRP during anti PD-(L)1 treatment may harbor more information for predicting progression risk than a single CRP level prior treatment initiation. Recent biostatistical innovation has brought forward so-called joint models of longitudinal and time-to-event data which are well suited for quantifying links between a biomarker trajectory and a clinical outcome [31]. In this observational bi-center study we aim to provide a comprehensive investigation and comparison of pretreatment and longitudinal CRP levels as potential predictive biomarkers of treatment response, disease progression and death in patients with advanced NSCLC undergoing anti PD-1 or anti PD-L1 treatment.

## 2. Methods

### 2.1. Study Concept

In this retrospective bi-center study, we aimed to (1) assess the prognostic association between CRP levels at ICI treatment initiation and ICI treatment outcomes within a single-center cohort (“Graz cohort”), (2) confirm the findings in an external validation cohort (“Nuremberg cohort”) and (3) explore the longitudinal evolution of CRP during ICI therapy and the potential impact of dynamic CRP changes on ICI treatment outcomes in the “Graz cohort.” Treatment outcomes of primary interest were progression-free survival (PFS) and overall survival (OS), while the objective response rate (ORR) was a secondary outcome, respectively. The study was approved by the IRB of the medical university of Graz (26–196 ex 13/14) and by the ethics committee of the Friedrich Alexander University Erlangen, Nuremberg, Germany (Nr. 62_17 B). Written informed consent was not obtained from individual patients, because this is not mandated in Austria and Germany for retrospective database studies given approval by an ethics committee. All data generated or analyzed during this study are included in this published article (and its Appendix A). Statistical analysis code is available on request from the corresponding author. The dataset analyzed during the current study cannot be shared under the current ethics committee approval.

### 2.2. Study Design—Graz Cohort (Discovery Cohort)

We retrospectively included all adult patients (i.e., ≥18 years) with histologically confirmed metastatic and/or recurrent and/or unresectable NSCLC who received at least one dose of an ICI targeting the PD-1/PD-L1 axis at the Medical University of Graz Hospital (Division of Oncology & Division of Pulmonology, both Department of Internal Medicine) between May 2015 and December 2018. Patients were identified by centralized records of the in-house oncology pharmacy, thus obtaining 100% local coverage. Baseline data on patient demographics, tumor characteristics, all subsequent anti-neoplastic treatment lines, and clinical outcomes were ascertained from our in-house electronic healthcare database as previously described, from written progress notes of the local Department, and from the central registry of the Austrian social security providers association (for all-cause death) [32,33]. Pre-specified co-primary analyses were (1) the relationship between CRP levels at baseline, PFS and OS, and (2) the relationship between the CRP trajectory and PFS, as assessed with joint models for longitudinal and time-to-event data. Secondary analysis was the relationships between CRP levels at baseline and the objective response rate (ORR) according to assessment of radiographic response by treating physicians in analogy to immune-related response evaluation criteria in solid tumors (irRECIST). The start date for time-to-event analyses was defined as the day of ICI treatment initiation. PFS was defined as the time from the start date until disease progression, censoring alive or death, whatever came first. OS was defined as the time from the start date until censoring alive or death-from-any-cause. Both PFS and OS analyses were truncated at two years of follow-up. Patients who were not evaluable for radiographic response (NE) were excluded from the ORR analyses. The CRP measurement within 14 days prior to the start date being closest to the start date was considered as the baseline CRP, and all CRP measurements from 14 days prior the start date until the PFS or OS end dates were considered for the joint modeling analysis.

### 2.3. External Validation Cohort

One-hundred-eleven patients with metastatic NSCLC who started ICI treatment for metastatic NSCLC between 03/2016 and 02/2019 at the Lung Cancer Center Nuremberg, Germany, served as the validation group. All patients received at least one dose of an ICI targeting the PD-1 axis in palliative 1st (pembrolizumab) or 2nd line (nivolumab) treatment. Patient data were documented by the medical records and electronically for the prospective biomarker LUISE study, where liquid biopsies from blood are taken before treatment and during ICI therapy till progression or death. Baseline data on patient demographics, tumor characteristics, all subsequent anti-neoplastic treatment lines, side effects, follow-up and clinical outcomes were available for all patients. PD-L1 expression analysis by immunohistochemistry was only necessary for approval of pembrolizumab, but not for nivolumab. Therefore, PD-L1 expression data are missing in many patients treated by nivolumab. Data of all-cause mortality were available from the clinical records and the registry offices in Germany. Actual data for baseline CRP (not longer than 14 days before start of ICI treatment) were missing for 10 patients. Thus, data from 101 patients could be used for external validation.

### 2.4. Laboratory Analysis Graz

For CRP measurement blood was drawn by sterile antecubital venipuncture into primary sample tubes containing lithium–heparin and separator gel (Vacuette^®^ by Greiner Bio-One, Kremsmünster, Austria). CRP levels were assessed in plasma (centrifugation at 2300× *g* for 10 min) using a certified latex particle-enhanced immunologic turbidimetric assay ("CRPL3” on a cobas^®^ 8000 c701 analyzer by Roche Diagnostics). The validated limits of quantification are 0.3 and 350 mg/L, the clinical reference cutoff is set at 5 mg/L.

### 2.5. Laboratory Analysis Nuremberg

For CRP measurements we used blood drawn by sterile antecubital venipuncture into primary sample tubes containing lithium–heparin and separator gel (S-Monovette^®^ LH gel, Sarstedt, Nümbrecht, Germany). The determination of CRP was performed in plasma (centrifugation at 2500 *g* for 10 min; from June 2017 at 2500 *g* for 15 min) using a certified latex particle-enhanced immunologic turbidimetric assay (CRPL3 on a cobas ^®^ 8000 analyzer with a c702 module by Roche Diagnostics, Mannheim, Germany). The measuring range was from 0.3 to 350 mg/L, the clinical reference cutoff is set at 5 mg/L. Only results ≥5 mg/L were reported.

### 2.6. Statistical Methods

All statistical analyses were performed with Stata 15.0 (Stata Corp., Houston, TX, USA). Continuous variables were reported as medians (25th–75th percentile), and count data as absolute frequencies (%). Median follow-up time was estimated with the reverse Kaplan–Meier estimator [34]. The association between CRP levels at baseline and the clinical outcomes under study were evaluated with logistic regression, Kaplan–Meier estimators, log-rank tests and Cox proportional hazards regression. In all Cox regressions, we evaluated the proportional hazards using Schoenfeld tests. In case of evidence for a violation of the PH assumption, we used flexible parametric models on the log (cumulative hazard) scale fully allowing for time-dependent effects (stata routine stpm2). For external validation, we re-performed these analyses in the Nuremberg cohort. The change in CRP over time was analyzed with linear mixed models, while we used joint models of longitudinal and time-to-event data to quantify the relationship between CRP trajectories and clinical outcomes. Briefly, joint models consist of a longitudinal component (in this study: the CRP trajectory) and a “survival” component (in this study: PFS and OS) that are linked to each other via the association parameter α. The joint model was specified as follows: (1) Linear mixed model with random intercept for the longitudinal component, (2) Weibull proportional hazards model for the “survival” component of the model, (3) “current association” specification of the association parameter α, [35] and (4) an unstructured variance–covariance matrix. The final follow-up time specification for the longitudinal component (linear, quadratic, and/or cubic) was selected based on the lowest value of Aikaike’s information criterion (AIC). CRP trajectories for patients who did and did not develop disease progression or death were graphed with margins plots and nonparametric smoothers. Moreover, a sensitivity analyses was performed with a “1st derivative” specification of α (i.e., the “slope” or rate of CRP change per year) [35]. Patient-specific outcome predictions according to the CRP trajectory were obtained with code contributed by MJC (stata routine stjmcsurv (in development)) [35]. In an exploratory analysis, we considered the relative change in CRP from the start date to the highest/lowest CRP value within 8 weeks after the start date as a predictor of clinical outcomes. Missing data are reported in Table 1, and a complete case analysis was performed.

## 3. Results

### 3.1. Analysis at Baseline and Crude Outcome Rates (Graz Cohort)

Ninety patients with a median age of 67 years (25th–75th percentile: 59–74) were included in the analysis, of whom *n* = 44 (49%) were female and *n* = 65 (72%) had adenocarcinomas (Table 1). Approximately one third of the patients (*n* = 32 (36%) received treatment in curative intent before they developed advanced/metastatic disease with subsequent ICI initiation. Most patients (*n* = 82, 91%) received ICIs during the 1st or 2nd line of systemic therapy, and all patients received ICI monotherapy only. During ICI treatment, the radiographic ORR as assessed by treating physicians was 20% (95% CI: 12–30), including 2 complete and 16 partial remissions and 16 patients, who were not evaluable for radiographic response assessment (Appendix A). Patients were followed-up for a median interval of 21.7 months (range: 5 days–3.6 years). Seventy-five and 25 percent of the cohort were followed for at least 11.5 and 29.8 months, respectively. During this follow-up period, we observed 49 disease progressions (median PFS = 4.8 months, 95% CI: 3.1–7.6), and 57 patients died (median OS = 7.7 months, 95% CI: 4.7–17.8, Appendix A).

On average, patients who developed progressive disease and/or died during follow-up did not differ significantly in any baseline characteristic from patients who remained alive and free from disease progression with the numbers of patients we had (Table 1). Nonetheless, patients with progressive disease and/or death during follow-up tended to have lower PD-L1 expression and higher baseline CRP levels. Baseline CRP was missing in *n* = 5 patients. Patients with high baseline CRP, as defined by an empirical cutoff at the 75th percentile of the baseline CRP distribution (CRP > 66.1 mg/L, *n* = 21), had a significantly lower prevalence of secondary primary malignancies, a more frequent history of prior adjuvant chemotherapy and received a significantly lower number of ICI therapy cycles. Other covariables very similarly distributed between patients above and below the CRP cutoff at the 75th percentile (Appendix A).

### 3.2. Elevated Baseline CRP Predicts Poor Response to ICI Therapy (Graz Cohort)

Seventy-three patients (81%) had a baseline CRP measurement and their radiographic response evaluated. Among this population, patients with higher CRP levels at baseline featured significantly lower odds of objective response to ICI therapy (odds ratio (OR) of objective response per doubling of baseline CRP = 0.68, 95% CI: 0.51–0.92, *p* = 0.013). In detail, ORR estimates were 45% (95% CI: 23–68), 25% (9–49), 10% (1–32) and 8% (0–36) in patients with a baseline CRP within the first (CRP ≤ 7.7 mg/L), second (CRP > 7.7 mg/L but ≤21.6 mg/L), third (CRP > 21.6 mg/L, but ≤66.1 mg/L) and fourth (CRP > 66.1 mg/L) quartile of the baseline CRP distribution, respectively (Fisher’s exact *p* = 0.033, Figure 1). Among all covariables, only lower age emerged as a statistically significant univariable predictor of worse ORR (Table 2). The association between elevated baseline CRP and lower odds of response prevailed in multivariable analysis adjusting for age (adjusted OR per doubling of baseline CRP = 0.66, 95% CI: 0.47–0.91, *p* = 0.011), as well as in analyses adjusting for LDH, NLR or the LIPI score (Appendix A).

Data are only for patients with observed baseline CRP and a response assessment other than NE (not evaluated). Thus, numbers of patients in the CRP quartiles are not balanced. CRP quartile cutoffs were as follows: Q1: CRP ≤ 7.7 mg/L, Q2: CRP > 7.7 mg/L, but ≤21.6 mg/L, Q3: CRP > 21.6 mg/L, but ≤66.1 mg/L and Q4: CRP > 66.1 mg/L. Abbreviations: ORR—objective response rate; CRP—C-reactive protein; Q—quartile.

### 3.3. Elevated Baseline CRP Predicts Disease Progression and Death (Graz Cohort)

Both PFS (hazard ratio (HR) per doubling of baseline CRP = 1.43, 95% CI: 1.22–1.69, *p* < 0.0001, Figure 2A, OS (HR per doubling of baseline CRP = 1.35, 95% CI: 1.15–1.57, *p* < 0.0001, Figure 2B) were significantly worse in patients with elevated baseline CRP, respectively. No evidence for a violation of the proportional hazard assumption was observed for CRP in PFS analysis (Global Schoenfeld test for continuous CRP model: *p* = 0.363) and OS analysis (*p* = 0.079), respectively. Other univariable predictors of a worse PFS and/or OS included, among others, higher age, male gender, stage IV at initial diagnosis/no prior treatment in curative intent, ICI treatment in 2nd- rather than 1st-line, higher NLR, higher LDH and higher LIPI scores, respectively (Table 2). The associations between elevated CRP and worse PFS (HR per doubling of baseline CRP = 1.37, 95% CI: 1.16–1.63, *p* < 0.0001) and OS (HR per doubling of baseline CRP = 1.42, 95% CI: 1.18–1.71, *p* < 0.0001) were independent of these prognostic factors, respectively (Appendix A). Moreover, results were independent of other established outcome biomarkers in NSCLC immunotherapy, including LDH, NLR and the LIPI (Appendix A).

### 3.4. Baseline CRP is Confirmed as a Predictor of Adverse Progression and Mortality Outcomes But Not Anti PD-(L)1 Treatment Response in External Validation (Nuremberg Cohort)

For external validation, we used the same CRP cut-offs as in the Graz cohort. The external validation cohort of 101 NSCLC patients undergoing ICI therapy was similar to the primary dataset (Graz cohort) in terms of the CRP distribution (median CRP: 32 mg/mL (25th–75th percentile: 11–64), follow-up interval (median: 19.1 months (12.0–26.7)) and several other baseline covariables (Appendix A). Otherwise, average PD-L1 expression was slightly higher, ORR was higher (Appendix A), and some covariables were not available in the Nuremberg cohort, respectively. Elevated CRP was not confirmed as a predictor of ICI response in the Nuremberg cohort (OR per doubling of baseline CRP = 0.97, 95% CI: 0.75–1.24, *p* = 0.784, Table 3, Appendix A). On the other hand, the external validation could fully confirm the prognostic association between elevated baseline CRP and increased all-cause mortality (Table 3, Figure 3B). In terms of PFS, strong evidence for a violation of the proportional hazard assumption was observed (Schoenfeld test for continuous CRP model *p* = 0.008, crossing curves in Figure 3A). In detail, an interaction between baseline CRP and linear follow-up time suggested that the “effect” of CRP on PFS (HR per doubling of baseline CRP = 1.41, 95% CI: 1.13–1.76, *p* = 0.002) was strongest early after ICI treatment initiation, but subsequently became weaker during follow-up, with a 6-fold multiplicative reduction in the hazard ratio per each year of time after ICI treatment initiation (HR for interaction between CRP and follow-up time = 0.57, 95% CI: 0.38–0.86, *p* = 0.008). Thus, a flexible parametric model was used. In this regression analysis fully taking into account the time-dependency, higher baseline CRP could be validated as a predictor for worse PFS as a continuous variable, whereas the prognostic association of CRP as a quartile variable with the Graz cohort cutoffs was only borderline statistically significant (Table 3).

### 3.5. Analysis of Longitudinal CRP Trajectories During Anti PD-(L)1 Therapy and progression (Graz Cohort)

To investigate the longitudinal evolution of CRP under anti PD-(L)1 treatment in the Graz cohort, we studied 1150 CRP measurements from baseline until the development of a PFS event or censoring alive without such an event (Average n of CRP measurements per patient: 13, range 1–54). In patients with a PFS event, average longitudinal CRP levels remained relatively stable early on and then subsequently increased, while they slightly declined early on and then remained relatively constant in patients without a PFS event (Appendix A). Looking at individual CRP trajectories using spaghetti plots with an inverted time axis and nonparametric smoothing, we observed a high within-patient variation of CRP over time, increasing CRP levels over time, and a stronger increase in CRP levels over time in patients who developed a PFS event than patients who remained free from a PFS event that was particularly pronounced before PFS onset (Figure 4). In univariable joint modeling of CRP trajectories and time-to-PFS, an elevated CRP trajectory over time was associated with a higher risk of developing the PFS event (hazard ratio per 10-mg/L increase in CRP = 1.17, 95% CI: 1.12–1.21, *p* < 0.0001). This association prevailed after multivariable adjustment for age and stage IV at initial diagnosis (Adjusted HR per 10-mg/L increase in CRP = 1.22, 95% CI: 1.15–1.30, *p* < 0.0001). Notably, when analyzing a “1st derivative” specification of the trajectory, a higher rate of CRP increase over time was also strongly prognostic for an increased risk of developing a PFS event. In detail, a 10 mg/mL/month faster increase in CRP levels over time predicted for 13-fold higher risk of experiencing a PFS event (HR = 13.26, 95% CI: 1.14–154.54, *p* = 0.039). Based on a “1st derivative” joint model for CRP and PFS adjusted for age and stage IV disease at initial diagnosis, patient-specific CRP trajectories could be used to obtain highly personalized “dynamic” predictions of a PFS event. This concept is illustrated in Figure 5 according to two patients of the Graz cohort, with the red dash-dotted line representing their last visit within the study.

### 3.6. Early CRP Decline after ICI Initiation Predicts ICI Therapy PFS (Graz Cohort)

In an exploratory, hypothesis-generating analysis we investigated the maximum change in CRP during the first 8 weeks after ICI initiation as a potential predictor for PFS (in those 74 patients with pertinent data available). While increasing CRP was not associated with higher risk of a PFS event, decreasing CRP during this early treatment period (as defined by the maximum percent decrease in CRP than the baseline value) was strongly associated with a favorable PFS experience (Figure 6). In detail, each 10% reduction in CRP predicted for a 0.9-fold reduction in PFS risk (HR = 0.91, 95% CI: 0.83–0.99, *p* = 0.036).

## 4. Discussion

In this bi-center observational cohort study we comprehensively investigated CRP as a treatment response and disease outcome biomarker in patients with advanced NSCLC undergoing anti PD-(L)1 inhibitor immunotherapy. We identified and externally validated pretreatment CRP as a marker of poor PFS and OS. Based on longitudinal CRP trajectory analysis incorporating 1150 CRP measurements, personalized dynamic predictions of progression risk could be obtained. Early CRP decline emerged as a strong predictor of favorable outcome, whereas elevated CRP trajectories were independently associated with higher progression risk. In summary, these data support the concept that CRP should be explored by future prospective studies as a simple non-invasive biomarker for assessing and monitoring treatment benefit during anti PD-(L)1 treatment in advanced NSCLC.

### 4.1. Pretreatment CRP Levels

First, we investigated the prognostic and predictive impact of pretreatment CRP levels. We found that elevated pretreatment CRP levels were associated with poor disease outcome as indicated with shortened PFS and OS. This is in line with previous studies investigating the prognostic impact of pretreatment CRP levels in cancer patients treated with ICI [27,28,29,36,37]. Suzuki et al. showed a strong association between elevated pretreatment CRP levels and worse OS in patients with metastatic renal cell carcinoma treated with nivolumab. They further showed that a decline of CRP ≥ 25% during ICI treatment was predictive of better treatment response [38]. In a retrospective cohort study including 124 NSCLC patients treated with the PD-1 inhibitor nivolumab Oya et al. identified elevated CRP levels above the upper limit of normal (≥1.0 mg/dL = 10 mg/L) as an independent predictor of decreased PFS, whereas for OS and ORR analysis only a univariate model was performed [29]. Interestingly, in our patient cohorts median baseline CRP levels were markedly higher. Naquash et al. defined CRP > 50 mg/L as an optimal cut off and found that elevated levels of CRP were associated with decreased OS. However, no information regarding the association of CRP levels with PFS and ORR were provided [28]. Another recently published retrospective study by Livanainen et al. investigated the prognostic role of pretreatment CRP levels in a mixed cohort of NSCLC, metastatic melanoma, renal and bladder cancer patients treated with anti- PD-(L)-1 agents. In the subgroup of NSCLC consisting of 16 patients in the discovery cohort and 42 patients in the validation cohort pretreatment CRP levels above 10 mg/mL were associated with shortened PFS and OS in univariate analysis, however no multivariable model was performed [27]. In our study we aimed to provide a thorough investigation of the prognostic and predictive value of pretreatment CRP level and its association with PFS, OS and the ORR. For that purpose, we performed continuous and categorical biomarker analysis and aimed to validate our findings in an external cohort, which was similar to the discovery cohort. We found that patients with elevated CRP levels had significantly shorter median PFS and OS. In detail, a doubling of pretreatment CRP level was associated with a 1.4 higher relative risk of disease progression or death. For categorizing patients into different risk groups, CRP cut offs according to the quartiles of the overall CRP distribution in the Graz cohort were defined. (cut offs: Q1: CRP ≤ 7.7 mg/L, Q2: CRP > 7.7 mg/L but ≤21.6 mg/L, Q3: CRP > 21.6 mg/L but ≤66.1 mg/L and Q4: CRP > 66.1 mg/L)Particularly patients with highly elevated CRP levels had very poor outcome with a 5-fold higher risk of progression or death. We could fully confirm these findings in our external validation cohort. In contrast, we could not validate the strong association of pretreatment CRP levels with ORR to anti PD-(L)1 agents found in the Graz cohort. This underlines the use of pretreatment CRP levels as a prognostic, but not predictive biomarker. Notably, in the Graz cohort, prognostic associations between CRP and oncologic outcomes were independent of NLR, LDH, and the LIPI score, a new validated risk model for adverse ICI therapy outcomes in NSCLC consisting of NLR and LDH [39].

### 4.2. CRP Trajectories

Although useful for primary prognostic risk stratification, single pretreatment biomarker measurements often may have limited predictive value and may not fully reflect the complex course of cancer. We hypothesized that longitudinal repetitive CRP measurements and their trajectories during anti-PD(L)1 treatment may correlate with disease activity and thus may harbor additional potential for monitoring treatment response and prediction of disease progression. Simeone et al. found that decreasing levels of CRP from baseline to week 12 significantly correlated with better disease control rate and longer overall survival in metastatic melanoma patients treated with the anti- CTLA4 agent ipilimumab [37]. Recently, Schiwitza et al. showed that a weighted score incorporating relative changes of three laboratory biomarkers including CRP has a high sensitivity in the prediction of treatment benefit to nivolumab in NSCLC patients [40]. In the present study, we applied the recently developed joint model analysis approach, which represents a powerful biostatistical tool for assessing the association between a longitudinal biomarker trajectory and a time-to-event outcome such as disease progression. Based on the incorporation of longitudinal biomarker trajectories within each patient joint models allow to provide individualized outcome predictions at each timepoint a new measurement of the respective marker is entered into the model [31,41,42,43]. For this purpose, we analyzed 1150 CRP measurements obtained from 90 patients during anti-PD(L)-1 treatment. We found that overall CRP levels increase over time, which is in line with previous studies indicating that rising CRP levels correlate with disease progression [14]. In addition, we could establish a clear link between CRP trajectories and progression risk that was independent of other prognostic factors. In detail, an elevated trajectory of CRP over time, as well as a faster increase in CRP was significantly associated with an increased progression risk. Importantly, we could demonstrate that joint model analysis of patient specific CRP trajectories offers great potential for individualized prediction of progression risk, a concept that is illustrated in Figure 4. Finally, we aimed to investigate whether CRP trajectories during the first eight weeks of anti PD-(L)1 treatment could be used for early prediction of treatment response. We found that an early decline of CRP is highly indicative of a lower progression risk, which is in line with previous studies reporting a strong association between declining CRP levels and better outcome in patients undergoing chemotherapy [44,45,46]. The decline of CRP in patients who respond to antineoplastic treatment may be related to a decreased secretion of tumor derived proinflammatory cytokines coming along with tumor shrinkage. However, further studies are needed in order to gain a better mechanistic understanding of the complex interplay between cancer and inflammation.

In summary, these findings support the use of longitudinal CRP level assessment as a cheap, simple and readily available tool that helps to predict disease outcome and response to anti-PD(L1) therapy.

Several limitations of our study must be discussed. First, the retrospective nature of this study needs to be taken into account regarding issues surrounding selection and information bias. Second, no information regarding potential time-dependent confounders such as infections or hepatic dysfunction, that may have affected CRP levels were available and could thus not be included in the analysis. Third, no control group of patients treated with chemotherapy was available. Fourth, no longitudinal CRP levels from the external validation cohort were available. Despite these limitations, this is the largest study that provides a comprehensive investigation of the prognostic and predictive value of pretreatment and longitudinal CRP levels in this setting.

## 5. Conclusions

We found that elevated pretreatment CRP levels are associated with poor disease outcome making them useful prognostic markers for primary risk stratification. On the other hand, trajectories of longitudinal CRP measurements during anti PD(L)-1 treatment harbor important information on progression risk and treatment response. An elevated and fast increase of CRP over time is a strong indicator of an elevated progression risk, whereas an early decline of CRP is associated with better treatment response. The joint model analysis approach represents a powerful statistical tool for individualized risk prediction. In conclusion, this study shows that CRP may serve as a simple biomarker for assessing and monitoring ICI treatment benefit in advanced NSCLC patients.

## Figures and Tables

**Figure 1 cancers-12-02319-f001:**
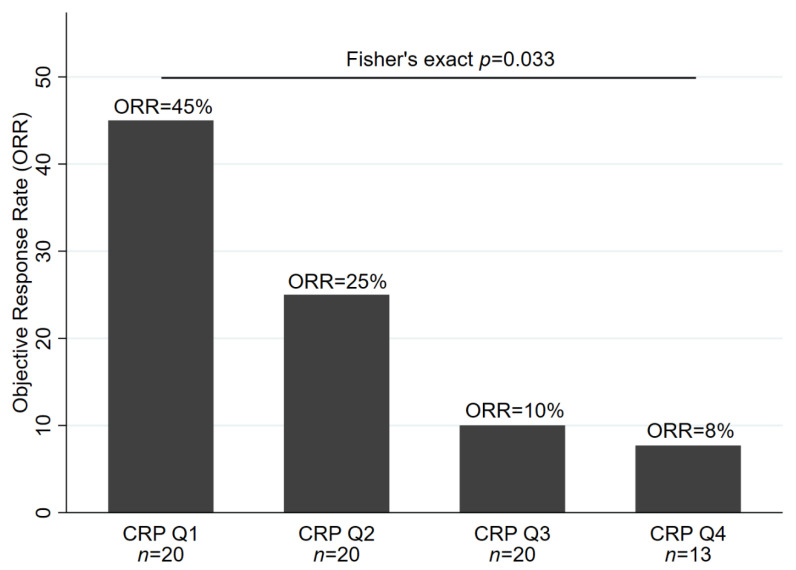
Baseline C-reactive Protein (CRP) and physician-assessed radiographic objective response rate in the Graz cohort (*n* = 73).

**Figure 2 cancers-12-02319-f002:**
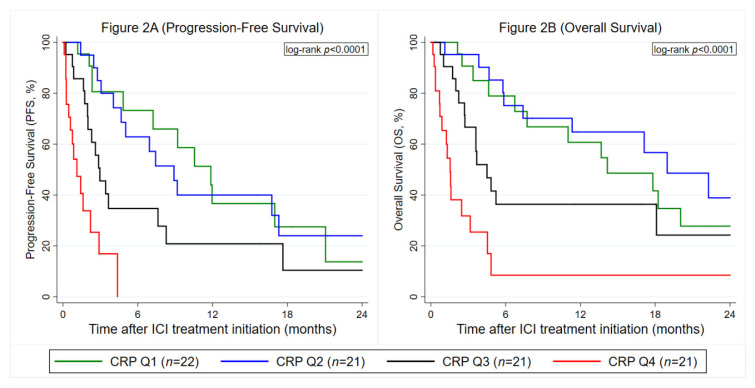
Prognostic association between baseline CRP levels and immune checkpoint inhibitors (ICI) treatment outcomes in the Graz cohort (*n* = 85). (**A**) Progression-Free Survival. (**B**) Overall Survival. Data are only for patients with observed baseline CRP. Curves were estimated with Kaplan–Meier estimators. The left panel (Figure 2A) depicts a PFS analysis and the right panel (Figure 2B) an OS analysis, respectively. CRP cutoffs were as follows: Q1: CRP ≤ 7.7 mg/L, Q2: CRP > 7.7 mg/L, but ≤21.6 mg/L, Q3: CRP > 21.6 mg/L, but ≤66.1 mg/L and Q4: CRP > 66.1 mg/L. Abbreviations: PFS—progression-free survival; OS—overall survival; CRP—C-reactive protein; Q—quartile; ICI—immune checkpoint inhibitor.

**Figure 3 cancers-12-02319-f003:**
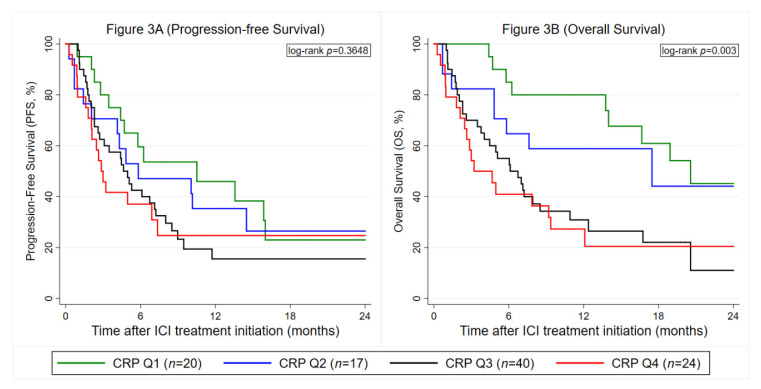
External validation of the prognostic association between baseline CRP levels and ICI treatment outcomes in the Nuremberg cohort (*n* = 101). (**A**) Progression-Free Survival. (**B**) Overall Survival. Data are only for patients with observed baseline CRP. Curves were estimated with Kaplan–Meier estimators. The left panel (Figure 2A) depicts a PFS analysis and the right panel (Figure 2B) an OS analysis, respectively. CRP cutoffs were taken from the Graz cohort and were as follows: Q1: CRP ≤ 7.7 mg/L, Q2: CRP > 7.7 mg/L, but ≤21.6 mg/L, Q3: CRP > 21.6 mg/L, but ≤66.1 mg/L and Q4: CRP > 66.1 mg/L. Abbreviations: PFS—progression-free survival; OS—overall survival; CRP—C-reactive protein; Q—quartile; ICI—immune checkpoint inhibitor.

**Figure 4 cancers-12-02319-f004:**
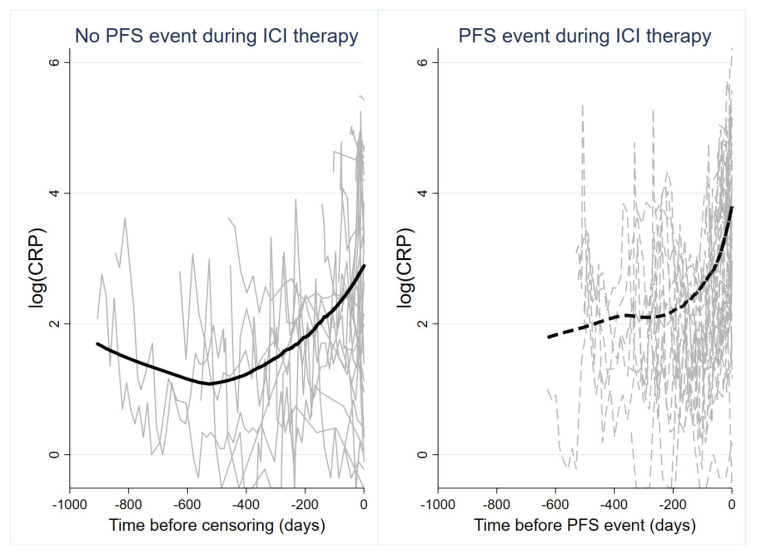
Line plot of CRP trajectories in patients who did (right panel, gray dashed lines) and did not (left panel, gray solid lines) develop a PFS event during ICI therapy. Each line represents the log(CRP) trajectory of a single patient. The bold solid line (left panel) and bold dashed line (right panel) represent moving averages (locally weighted sum-of-squares (LOWESS) nonparametric smoother). Note that the time on the x-axis of both panels is inverted, i.e., it represents the time before a PFS event or censoring without a PFS event. Abbreviations: PFS—progression-free survival; ICI—immune checkpoint inhibitor.

**Figure 5 cancers-12-02319-f005:**
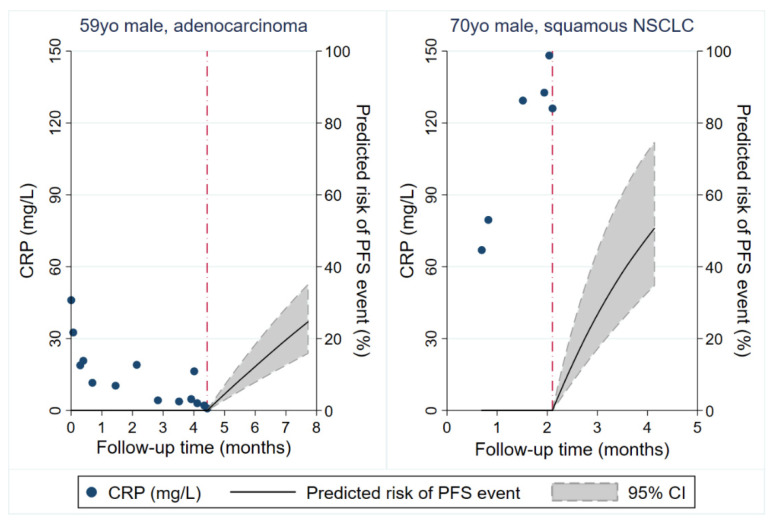
Personalized prediction of ICI therapy PFS outcomes based on individual patients longitudinal CRP trajectories. Predictions were obtained from a joint model of longitudinal and time-to-event data. Figure depicts predictions of 3-month risks of developing a PFS event for two study patients. The left panel depicts a 59-year-old male with synchronously metastasized adenocarcinoma (including brain metastases) and significant comorbidity who had decreasing CRP levels during 1st-line therapy with pembrolizumab. His predicted risk of a PFS event 3 months after his last study visit is 25%. Computed tomography staging examinations one week after his last study visit revealed stable disease, and the patient continued ICI therapy. The right panel depicts a 70-year-old male in limited performance status (ECOG 2) with synchronously metastasized squamous NSCLC and steadily increasing CRP levels during 1st-line therapy with pembrolizumab. His predicted risk of a PFS event 3 months after his last study visit is above 50%. The patient was referred to mobile palliative care services shortly after the last study visit due to symptomatic pleural effusion and radiographically documented progressive disease. Red dash-dotted line: Last study visit. Abbreviations: yo—year-old; NSCLC—non-small cell lung cancer; CRP—C-reactive protein; PFS—progression-free survival.

**Figure 6 cancers-12-02319-f006:**
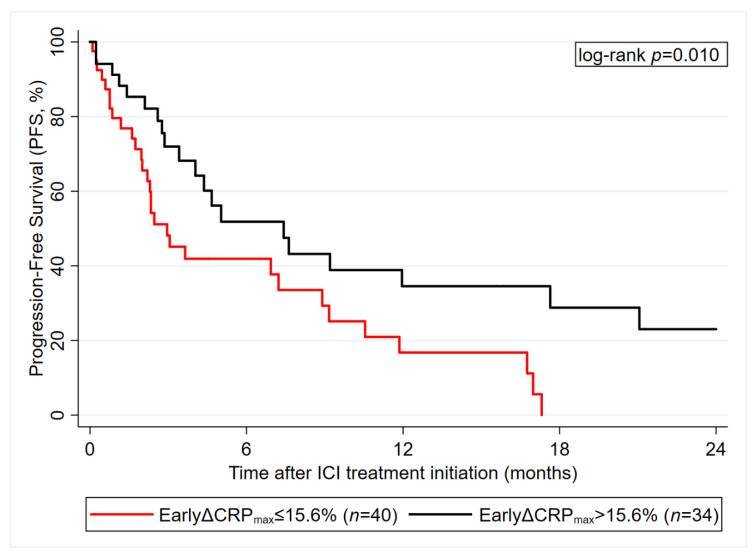
Exploratory, hypothesis-generating analysis of “early CRP decline” as a predictor of ICI therapy PFS outcome in the Graz cohort (*n* = 74). Data are from patients with observed baseline CRP and available 8-week CRP data. Early CRP decline was defined as the maximum percent decline in CRP during the first 8 weeks after ICI treatment initiation, with patients with CRP increase set to 0. A cutoff at 15.6% was derived using Youden’s Index. Abbreviations: PFS—progression-free survival; ICI—immune checkpoint inhibitor; CRP—C-reactive protein; EarlyΔCRP_max_—early relative decline in CRP.

**Table 1 cancers-12-02319-t001:** Baseline characteristics of the Graz cohort—distribution overall and by progression-free survival (PFS) event status.

Variable	n (% miss.)	Overall (*n* = 90)	No Progression or Death (*n* = 31)	Progression or Death (*n* = 59)	*p* *
**Demographic characteristics**					
Age at ICI initiation (years)	90 (0%)	67 (59–74)	69 (57–76)	66 (60–73)	0.527
Female gender	90 (0%)	44 (49%)	11 (35%)	33 (56%)	0.065
BMI at ICI initiation (kg/m²)	83 (8%)	24.3 (20.7–27.3)	25.2 (21.2–29.7)	24.2 (20.5–27.2)	0.391
Charlson comorbidity index at ICI initiation (points)	90 (0%)	8 (5–9)	8 (5–10)	8 (5–9)	0.317
Past or present smoker	86 (4%)	67 (78%)	23 (77%)	44 (79%)	0.839
ECOG at ICI initiation (points)	59 (34%)	0 (0–1)	1 (0–1)	0 (0–1)	0.843
Second primary malignancy at any time	86 (4%)	18 (21%)	5 (17%)	13 (23%)	0.477
**Tumor variables**					
Adenocarcinoma	90 (0%)	65 (72%)	21 (68%)	44 (75%)	0.492
Stage IV at initial NSCLC diagnosis	90 (0%)	55 (61%)	16 (52%)	39 (66%)	0.180
EGFR mutation	73 (19%)	2 (3%)	1 (4%)	1 (2%)	0.999
EML4–ALK rearrangement	73 (19%)	1 (1%)	0 (0%)	1 (2%)	0.999
ROS1 overexpression	61 (32%)	0 (0%)	0 (0%)	0 (0%)	N/A
BRAF mutation	23 (74%)	2 (9%)	1 (8%)	1 (9%)	0.999
PD-L1 expression (%)	68 (24%)	45 (1–80)	60 (15–88)	15 (0–80)	0.087
**Treatment prior ICI**					
Primary treatment intent: curative **	90 (0%)	32 (36%)	14 (45%)	18 (31%)	0.168
---Any neoadjuvant therapy (RTx, CTx, RCTx)	32 (0%)	8 (25%)	4 (29%)	4 (22%)	0.681
---Any definitive RCTx	32 (0%)	6 (19%)	4 (29%)	2 (11%)	0.365
---Any curative surgery	32 (0%)	23 (72%)	9 (64%)	14 (78%)	0.400
---Any adjuvant therapy (CTx, RTx)	32 (0%)	12 (38%)	3 (21%)	9 (50%)	0.147
**ICI treatment variables**					
ICI treatment line	90 (0%)	/	/	/	0.170
---1st-line	/	38 (42%)	17 (55%)	21 (36%)	/
---2nd-line	/	44 (49%)	11 (35%)	33 (56%)	/
---3rd, 4th or 5th-line	/	8 (9%)	3 (10%)	5 (8%)	/
ICI agent	90 (0%)	/	/	/	0.176
---nivolumab	/	49 (54%)	12 (39%)	37 (63%)	/
---pembrolizumab	/	37 (41%)	17 (55%)	20 (34%)	/
---Atezolizumab	/	4 (4%)	2 (6%)	2 (4%)	/
ICI in more than 1 treatment line	90 (0%)	1 (1%)	0 (0%)	1 (2%)	0.999
Number of ICI cycles	81 (10%)	5 (3–15)	7 (3–21)	5 (2–14)	0.280
**Laboratory variables**					
CRP at baseline (mg/L)	85 (6%)	21.6 (7.7–66.1)	14.0 (5.9–65.3)	25.5 (8.7–72.6)	0.258
NLR at baseline (units)	82 (9%)	4.7 (3.1–8.9)	4.6 (3.2–7.6)	4.7 (3.0–9.2)	0.807
LDH at baseline (IU/L)	82 (9%)	263 (199–347)	251 (183–313)	270 (206–398)	0.240
LIPI score at baseline	81 (10%)	/	/	/	0.915
---0 points	/	22 (27%)	8 (30%)	14 (26%)	/
---1 point	/	36 (44%)	12 (44%)	24 (44%)	/
---2 points	/	23 (28%)	7 (26%)	16 (30%)	/
Overall number of CRP measurements	N/A	2090	595	1495	N/A
Number of CRP measurements per patient	90 (0%)	18 (9–31)	13 (7–26)	19 (11–32)	0.059
Average of all available CRP measurements (mg/L)	2090 (0%)	17.8 (5.1–60.9)	5.5 (1.8–16.5)	29.0 (8.7–73.2)	<0.0001

Data are medians (25th–75th percentile) for continuous data and absolute frequencies (%) for count data. n (% miss.) reports the number of patients with fully observed data for the respective variable (% missing). * *p*-values are from rank-sum tests, Fisher’s exact tests and χ^2^ tests, as appropriate. ** Variables in the section “Treatment prior ICI” are with *n* = 32 patients and the missingness percentage was consequently scaled to 100% for *n* = 32. Abbreviations: ICI—immune checkpoint inhibitor; BMI—body mass index; ECOG—Eastern cooperation oncology group performance status; NSCLC—non-small cell lung cancer; EGFR—epidermal growth factor receptor; EML4–ALK—Echinoderm microtubule associated protein-like 4 anaplastic lymphoma kinase; ROS1– ROS proto-oncogene 1; BRAF—v-Raf murine sarcoma viral oncogene homolog B; PD-L1—programmed death ligand 1; RTx—radiotherapy; CTx—chemotherapy; RCTx—chemoradiation; CRP—C-reactive protein; NLR—neutrophil lymphocyte ratio; LDH—lactate dehydrogenase; LIPI—lung immune prognostic index; N/A—not applicable.

**Table 2 cancers-12-02319-t002:** Univariable predictors of treatment outcome in the Graz cohort.

Variable	Outcome: Radiographic ORR (*n* = 73)		Outcome: PFS (*n* = 85)		Outcome: OS (*n* = 85)	
	OR	95% CI (*p*)	HR	95% CI (*p*)	HR	95% CI (*p*)
**Demographic characteristics**						
Age at ICI initiation (per 5 years increase)	1.46	1.04–2.05 (*p* = 0.030)	0.88	0.78–0.98 (*p* = 0.020)	0.87	0.78–0.97 (*p* = 0.013)
Female Gender	2.77	0.86–8.90 (*p* = 0.087)	0.83	0.49–1.42 (*p* = 0.495)	0.46	0.26–0.81 (*p* = 0.007)
BMI at ICI initiation (per 5 kg/m² increase)	1.56	0.88–2.77 (*p* = 0.127)	0.81	0.56–1.11 (*p* = 0.195)	0.87	0.64–1.19 (*p* = 0.380)
Charlson comorbidity index at ICI initiation (per 1 point increase)	1.12	0.93–1.36 (*p* = 0.241)	0.98	0.89–1.07 (*p* = 0.618)	0.99	0.90–1.09 (*p* = 0.871)
Past or present smoker	0.86	0.23–3.15 (*p* = 0.816)	1.35	0.69–2.63 (*p* = 0.379)	1.59	0.77–3.28 (*p* = 0.211)
ECOG at ICI initiation (per 1 point increase)	0.33	0.08–1.44 (*p* = 0.140)	1.87	0.96–3.65 (*p* = 0.064)	1.66	0.85–3.27 (*p* = 0.139)
Second primary malignancy at any time	0.64	0.16–2.60 (*p* = 0.535)	1.34	0.71–2.52 (*p* = 0.372)	1.21	0.63–2.32 (*p* = 0.575)
**Tumor variables**						
Adenocarcinoma	0.8	0.24–2.67 (*p* = 0.717)	1.15	0.63–2.08 (*p* = 0.653)	1.20	0.63–2.29 (*p* = 0.581)
Stage IV at initial NSCLC diagnosis	0.42	0.14–1.27 (*p* = 0.125)	1.78	1.02–3.10 (*p* = 0.044)	1.80	1.01–3.21 (*p* = 0.046)
PD-L1 expression (per 10% increase)	1.07	0.91–1.25 (*p* = 0.408)	0.98	0.90–1.06 *p* = 0.577)	0.98	0.90–1.07 (*p* = 0.672)
**Treatment prior ICI**						
Primary treatment intent: curative	1.08	0.36–3.27 (*p* = 0.889)	0.50	0.28–0.90 (*p* = 0.02)	0.49	0.27–0.91 (*p* = 0.023)
**ICI treatment variables**						
ICI treatment: 1st-line	Ref.	Ref.	Ref.	Ref.	Ref.	Ref.
---2nd-line	0.57	0.19–1.74 (*p* = 0.325)	1.61	0.91–2.86 (*p* = 0.100)	1.89	1.03–3.47 (*p* = 0.040)
---3rd, 4th or 5th-line	n/a	n/a	1.21	0.45–3.26 (*p* = 0.701)	1.56	0.61–4.00 (*p* = 0.358)
ICI agent	/	/	/			
---nivolumab	Ref.	Ref.	Ref.	Ref.	Ref	Ref.
---pembrolizumab	1.33	0.45–3.98 (*p* = 0.606)	0.98	0.56–1.72 (*p* = 0.943)	1.19	0.66–2.13 (*p* = 0.561)
---Atezolizumab	n/a	n/a	1.59	0.21–11.81 (*p* = 0.649)	2.41	0.32–18.09 (*p* = 0.394)
**Laboratory variables**						
NLR (per doubling)	0.76	0.43–1.34 (*p* = 0.337)	1.38	1.04–1.83 (*p* = 0.026)	1.51	1.14–2.02 (*p* = 0.005)
LDH (per doubling)	0.83	0.35–1.98 (*p* = 0.675)	1.51	1.04–2.21 (*p* = 0.032)	1.39	0.95–2.04 (*p* = 0.090)
LIPI: 0 points	Ref.	Ref.	Ref.	Ref.	Ref.	Ref.
---1point	1.17	0.33–4.19 (*p* = 0.805)	1.40	0.72–2.71 (*p* = 0.317)	1.45	0.71–2.98 (*p* = 0.313)
---2 points	0.38	0.06–2.23 (*p* = 0.281)	2.67	1.28–5.55 (*p* = 0.009)	3.38	1.54–7.43 (*p* = 0.002)

Results for the objective response rate are odds ratios (logistic regression), whereas results for progression-free and overall survival are hazard ratios (Cox regression), respectively. “Per doubling” coefficients were obtained by log2-transformation of the respective variable. Abbreviations: ORR—objective response rate; PFS—progression-free survival; OS—overall survival or odds ratio; HR—hazard ratio; 95% CI (*p*)—95% confidence interval (Wald test *p*-value); ICI—immune checkpoint inhibitor; BMI—body mass index; ECOG—Eastern cooperation oncology group performance status; NSCLC—non-small cell lung cancer; PD-L1—programmed death ligand 1; Ref—reference category; n/a—not applicable/estimable; NLR—neutrophil lymphocyte ratio; LDH—lactate dehydrogenase; LIPI—lung immune prognostic index.

**Table 3 cancers-12-02319-t003:** Prognostic associations between baseline CRP and clinical outcomes of ICI therapy in the Graz cohort (development cohort) and Nuremberg cohorts (external validation cohort).

		Graz Cohort		Nuremberg Cohort *	
Outcome	Variable	OR/HR	95% CI (*p*)	OR/HR	95% CI (*p*)
ORR	CRP (per doubling)	0.68	0.51–0.92 (*p* = 0.013)	0.97	0.75–1.24 (*p* = 0.784)
PFS	CRP (per doubling)	1.43	1.21–1.70 (*p* < 0.0001)	1.20	1.02–1.41 (*p* = 0.028)
	CRP: Q1 **	Ref.	Ref.	Ref.	Ref.
	CRP: Q2	1.04	0.47–2.28 (*p* = 0.921)	1.34	0.60–3.01 (*p* = 0.480)
	CRP: Q3	2.18	1.03–4.63 (*p* = 0.043)	2.04	0.96–4.33 (*p* = 0.064)
	CRP: Q4	8.46	3.62–19.77 (*p* < 0.0001)	2.28	0.95–5.44 (*p* = 0.064)
OS	CRP (per doubling)	1.38	1.17–1.64 (*p* < 0.0001)	1.30	1.12–1.51 (*p* = 0.001)
	CRP: Q1	Ref.	Ref.	Ref.	Ref.
	CRP: Q2	0.74	0.32–1.171 (*p* = 0.477)	1.41	0.54–3.68 (*p* = 0.476)
	CRP: Q3	1.70	0.78–3.69 (*p* = 0.179)	2.99	1.40–6.37 (*p* = 0.005)
	CRP: Q4	5.02	2.32–10.89 (*p* < 0.001)	3.51	1.56–7.88 (*p* = 0.002)

Data are odds ratios for the objective response rate and hazard ratios for progression-free and overall survival. CRP (per doubling) represents a log2-transformed CRP variable. CRP: Q1–CRP: Q4 represents a CRP variable of 4 quartile categories. * Flexible parametric models were used for the PFS analysis in the Nuremberg cohort due to a violation of the proportional hazard assumption. ** The same CRP quartile cutoffs were used in the Graz and Nuremberg cohort. These cutoff was obtained in the Graz cohort and are as follows: Q1: CRP ≤ 7.7 mg/L, Q2: CRP > 7.7 mg/L but ≤21.6 mg/L, Q3: CRP > 21.6 mg/L but ≤66.1 mg/L and Q4: CRP > 66.1 mg/L. Abbreviations: OR—odds ratio; HR—hazard ratio; 95% CI—95% confidence interval (Wald test *p*-value); ORR—objective response rate; CRP—C-reactive protein; Q—quartile; Ref—reference category; PFS—progression-free survival; OS—overall survival.

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
