# Peer review of "C-Reactive Protein (CRP) Levels in Immune Checkpoint Inhibitor Response and Progression in Advanced Non-Small Cell Lung Cancer: A Bi-Center Study"

_cancers, 2020, doi:10.3390/cancers12082319_

Round 1

Reviewer 1 Report

In the discussion, important to refer to articles related to LDH and Neutrophil: lymphocyte ratio. These are simple assessments available from standard blood tests in the clinic that potentially allow assessment of the likelihood of the patient to respond to treatment. High LDH is prognostic for a poor outcome as is a high neutrophil:lymphocyte ratio with combined scores, known as the LIPI score being strongly prognostic. These, like CRP, may reflect an immunosuppressive environment and a correlation between CRP and these markers would be interesting and add to the paper.

Other studies have shown that the clinical biomarkers LDH and Neutrophil:lymphocyte ratio are important in predicting resistance to immunotherapy. This needs to be discussed as CRP may simply reflect these. I asked the question also if the authors had this data and could they do a relationship to see if CRP is related to and adds to these markers.

Mezquita L, Auclin E, Ferrara R, Charrier M, Remon J, Planchard D, Ponce S, Ares LP, Leroy L, Audigier-Valette C, Felip E, Zerón-Medina J, Garrido P, Brosseau S, Zalcman G, Mazieres J, Caramela C, Lahmar J, Adam J, Chaput N, Soria JC, Besse B.Association of the Lung Immune Prognostic Index With Immune Checkpoint Inhibitor Outcomes in Patients With Advanced Non-Small Cell Lung Cancer. JAMA Oncol. 2018 Mar 1;4(3):351-357. doi: 10.1001/jamaoncol.2017.4771. PMID: 29327044

Reviewer 2 Report

Authors submitted manuscript titled ‘C-reactive protein (CRP) level in immune checkpoint inhibitors response and progression in advanced non-small-cell lung cancer: a bi-center study’ in which they evaluated importance of CRP in predicting response to immune checkpoint inhibitors in the treatment of non-small cell lung cancer (NSCLC). Authors performed retrospective study of patients from one center (Graz) and compared it to patients from validation center (Nuremberg). Authors evaluated association between baseline CRP levels and immune checkpoint inhibitors treatment outcomes and they also explored the longitudinal evolution of CRP during treatment of immune checkpoint inhibitors.

The results of the study have shown that pre-treatment elevated CRP levels were associated with worse PFS and worse OS, and lower ORR. Authors conclude that pre-treatment CRP and longitudinal follow-up of CRP levels can be predictors of response to immune checkpoint inhibitors.

I believe that this retrospective study can put further insights in trying to find predictive biomarkers of response to immune checkpoint inhibitors.

Thera are some minor suggestions to be changes:

  1. In the introduction section, more detailed explanation should be written about why increased CRP levels and inflammation could be worse prognostic biomarkers for efficacy of immune checkpoint inhibitor treatment. (I suggest moving it from discussion to introduction section)
  2. Explain in more details why pembrolizumab is referred as palliative treatment in 1st line, whereas nivolumab is just stated as second line
  3. Please explain in more details the cut-off values
  4. I suggest to include Supplementary figures 4 and 5 in the main text of the manuscript
  5. Discuss why decrease in CRP levels (and probably inflammation) could give benefit to immune checkpoint inhibitor treatment

Authors pointed out the limitations of the study fairly, but the study represents, to the best of my knowledge, the largest study that provides a comprehensive investigation of the prognostic and predictive value of pretreatment and longitudinal CRP levels in predicting the response to immune checkpoint inhibitors.

In conclusion,

My recommendation is to accept the manuscript for publication to the Cancers with minor changes suggested above.

Reviewer 3 Report

This a well-conducted study that increases the evidence and consolidates in multivariate analyses the correlation of high CRP levels with decreased PFS and OS, and identifies and association of higher CRP levels at baseline with lower odds of objective response to ICB. In addition, it generates new knowledge on the associations of the extent and speed of PCR increases over treatment time with disease progression, and of early declines with better treatment response.

Most of the limitations are indicated by the authors. In this regard, I would like to comment on the confounding factors increasing or decreasing CRP, related to infections or to hepatic function. Are there proxies of tumor-specific inflammation/immune response available in the clinical records, such as fluctuations in NK or PD1+ CD8+ cells?

With regards the predictive and prognosis potential, and given that the predictive potential in terms of ORR could not be validated in the Nuermberh cohort, would it be possible to integrate the CRP levels with the clinicopathological variables? As far as I understand, CRP is an independent prognostic predictor in multivariate analysis, but also other prognostic factors, that may be combined for increasing the predictive value.

Moeover, when it comes to the longitudinal joint model analysis approach, are these personalized predictions adding more information to the algorithms based on the clinicopathological characteristics of current use in the clinics? Do they predict better the risk of progression?

LDH has been reported as an ICB response biomarker in melanoma, and could be correlated to CRP in NSCLC. Is this variable in the patients’ records?

Finally, I am curious about toxicities to ICB in this scenario. Do you have any information in this regard?

Minor comments:

- Although PC-L1 was not significant in the univariate analysis in the Graz cohort, would it be possible to generate it for the Nuremberg cohort?

- In Table 1. Baseline characteristics of the Graz cohort - Distribution overall and by PFS event status, have the authors checked the distribution of the missing data among the two categories?

- Lines 205-208. High baseline CRP is associated with events that could be related to both a better and a worse prognosis. It is a bit confusing to me. Also, I do not relate the word “respectively” there.

- I wonder to which extent associating CRP with ORR as specific quartile cut-offs is useful for predicting response to ICB unless the values distributions are generally common in the population of patients with this disease.

- Would it be possible to add to the figures con time-to-event associations the number of patients per quartile and the events in each time point?

- Line 255-256: Should this sentence be in cursive? It seems another the tittle of another section.

- Supplementary figures are missing.

- The longitudinal data on the Nuremberg cohort would have been ideal to validate the model, although I consider it is still relevant. Could this prediction model be adjusted to longer follow up periods, and incorporate multiple variables registered in parallel over the course of the treatment?

- Line 393. It would be interesting to describe a bit more in detail the high sensitivity CRP biomarker of reference 38.

- Both “multivariate” and “multivariable” are used in the text I believe with similar meaning.

Round 2

Reviewer 1 Report

accept